# A Promising Intracellular Protein-Degradation Strategy: TRIMbody-Away Technique Based on Nanobody Fragment

**DOI:** 10.3390/biom11101512

**Published:** 2021-10-14

**Authors:** Gang Chen, Yu Kong, You Li, Ailing Huang, Chunyu Wang, Shanshan Zhou, Zhenlin Yang, Yanling Wu, Jianke Ren, Tianlei Ying

**Affiliations:** 1MOE/NHC/CAMS Key Laboratory of Medical Molecular Virology, School of Basic Medical Sciences, Shanghai Medical College, Fudan University, Shanghai 200032, China; 16111010053@fudan.edu.cn (G.C.); kongyu@fudan.edu.cn (Y.K.); zhonghualiyou@163.com (Y.L.); huang_ailing@fudan.edu.cn (A.H.); 15211010049@fudan.edu.cn (C.W.); shanshanzhou1110@163.com (S.Z.); 2Department of Pulmonary Medicine, Zhongshan Hospital, Fudan University, Shanghai 200032, China; yang_zhenlin@fudan.edu.cn; 3CAS Key Laboratory of Computational Biology, Shanghai Institute of Nutrition and Health, University of Chinese Academy of Sciences, Chinese Academy of Sciences, Shanghai 200031, China

**Keywords:** TRIM21, nanobody, TRIMbody, TRIM-away, TRIMbody-away, targeted protein degradation

## Abstract

Most recently, a technology termed TRIM-Away has allowed acute and rapid destruction of endogenous target proteins in cultured cells using specific antibodies and endogenous/exogenous tripartite motif 21 (TRIM21). However, the relatively large size of the full-size mAbs (150 kDa) results in correspondingly low tissue penetration and inaccessibility of some sterically hindered epitopes, which limits the target protein degradation. In addition, exogenous introduction of TRIM21 may cause side effects for treated cells. To tackle these limitations, we sought to replace full-size mAbs with the smaller format of antibodies, a nanobody (VHH, 15 kDa), and construct a new type of fusion protein named TRIMbody by fusing the nanobody and RBCC motif of TRIM21. Next, we introduced enhanced green fluorescent protein (EGFP) as a model substrate and generated αEGFP TRIMbody using a bispecific anti-EGFP (αEGFP) nanobody. Remarkably, inducible expression of αEGFP TRIMbody could specifically degrade intracellular EGFP in HEK293T cells in a time-dependent manner. By treating cells with inhibitors, we found that intracellular EGFP degradation by αEGFP TRIMbody relies on both ubiquitin–proteasome and autophagy–lysosome pathways. Taken together, these results suggested that TRIMbody-Away technology could be utilized to specifically degrade intracellular protein and could expand the potential applications of degrader technologies.

## 1. Introduction

Protein depletion or degradation technologies are widely used for researchers to understand the biological functions of intracellular proteins, which could be achieved by either interfering with protein synthesis or inducing protein degradation for controlling intracellular protein levels [1]. Traditional ways to disturb protein synthesis include manipulations of gene sequences by CRISPR/Cas9 genome editing technology, targeting mRNA transcripts by RNA interference (RNAi), and morpholino antisense oligonucleotides [2,3,4,5]. However, protein depletion by those approaches is indirect and depends on the inherent turnover of the protein, which may be time-consuming or result in depletion resistance for some long-lived proteins. To induce direct degradation of a protein of interest, a number of approaches have been designed harnessing the power and specificity of the intracellular protein degradation machinery, such as proteolysis-targeting chimaeras (PROTACs), lysosome-targeting chimaeras (LYTACs), dTAGs, chaperone-mediated autophagy targeting, and non-genetic IAP-dependent protein erasers (SNIPERs) [6,7,8,9,10]. All of these approaches have limitations, and their successful application depends on the protein of target and the experimental model of choice.

Recently, a technology termed TRIM-Away has been developed to acutely and rapidly degrade endogenous proteins in mammalian cells without change of the genome or mRNA expression level, using anti-target antibodies and TRIM21, which belongs to the TRIM (tripartite motif-containing) family [11,12]. The large majority of TRIM family proteins contain an N-terminal RBCC motif [13,14,15], followed by C-terminal domains of various length and diverse composition that are often used to target specific substrates and mediate diverse functions [16,17]. The structural arrangement of the RBCC motif is highly conserved within the TRIM protein family, while the C-terminal region is highly variable [18,19], of which the most common is the PRY-SPRY domain, also known as the B30.2 domain. TRIM21 has a mass of 54 kDa and consists of an N-terminal RBCC motif and a C-terminal PRY-SPRY domain [20,21]. The RBCC motif includes a RING domain with E3 ubiquitin ligase activity, a B-box domain, and a coiled-coil dimerization domain [22,23]. The PRY-SPRY domain binds to the Fc fragment of immunoglobulin with high affinity [24,25,26]. Therefore, TRIM-Away technology mainly relies on antibodies entering cells, where they recognize the target protein and bind to TRIM21, leading to substrate ubiquitination and degradation [27]. However, the conventional antibodies had relatively large size (150 kDa), which results in correspondingly low tissue penetration and inaccessibility of some sterically hindered epitopes [28,29], and limits the degradation efficacy of target protein in cells.

An attractive alternative is using smaller antibody fragments to replace full-size mAbs. Single-domain antibodies (sdAbs), also designated as VHHs or nanobodies, have a small size with only 15 kDa, resulting in unique advantages compared to mAbs, including larger number of accessible epitopes, relatively lower production costs, and improved biophysical properties [30,31]. Thus, we constructed a new type of fusion protein, designated as TRIMbody, by fusing the truncated form of TRIM21 with the nanobody. The truncated TRIM21 retained only the N-terminal RBCC domain and deleted the C-terminal PRY-SPRY domain. Therefore, TRIMbody possesses the functions of TRIM21 and mAbs, but has relatively small size. In this study, αEGFP TRIMbody, composed of a bispecific anti-EGFP (αEGFP) nanobody and truncated TRIM21, effectively degrade EGFP in cells that stably express EGFP protein in cytosol, whatever αEGFP TRIMbody is transient or inducible expressed. Moreover, we found both proteasome inhibitor and autophagy–lysosome inhibitor treatment could lessen the target protein degradation by αEGFP TRIMbody, indicating the TRIMbody function depends on both the proteasome and autophagy–lysosome pathways. Therefore, this TRIMbody-based protein degradation is designated as TRIMbody-Away technology, which could expand the landscape of the applications of degrader technologies and provide an alternative approach for potential therapeutic benefit in future.

## 2. Materials and Methods

### 2.1. Plasmids

HLTV-hTRIM21 (Addgene, Watertown, USA, 104973) and pHR-LaG16-LaG2 (Addgene, Watertown, USA, 85421) were purchased from Addgene. C-terminal His_6_-Flag tagged HLTV-αEGFP TRIMbody was generated by subcloning RBCC motif of TRIM21 and LaG16-LaG2 fragment into the HLTV expression vector (BamHⅠ-EcoRⅠ) using ClonExpress MultiS One Step Cloning Kit (Vazyme, Nanjing, China, C113-01). RBCC-LaG16-LaG2 (αEGFP TRIMbody) was then subcloned into a “all-in-one” tetracycline-inducible promoter construct (pTet-on-3G) using the BamHⅠ and EcoRⅠ sites. The LaG16-LaG2 gene was cloned into the pComb3x phagemid and the EGFP gene was cloned into the pET-28a as described above. pUg-EN2-EGFP was made by cloning the EGFP fragment into lentiviral vector pUg-EN2. All new constructs in this study were verified by DNA sequencing. The plasmids psPAX2 and pMD2.G were a kind gift from Shibo Jiang (Fudan University, Shanghai, China). The recombinant vector was transformed into Top10 or Stbl3 competent cell for propagation.

### 2.2. Expression and Purification of TRIM21 and TRIMbody

The recombinant vector plasmids were used for transformation of *E. coli* strain C43(DE3) pLysS cells. A single and freshly transformed colony was added to 4 mL 2× YT medium with 100 µg/mL ampicillin, 34 µg/mL chloromycetin, and 2% (wt/vol) glucose, incubated at 37 °C with vigorous shaking at 250 rpm for 3~4 h, and then transferred into 200 mL of SB medium with 100 µg/mL ampicillin for continued incubation until optical density of the culture at 600 nm reached 0.6~0.8 (after 3~4 h). Next, IPTG (isopropyl-1-thio-β-d-galactopyranoside) was added to a final concentration of 1 mM to induce protein expression, and the culture was further incubated overnight at 22 °C, 250 rpm. Bacteria were collected by centrifugation at 8000 rpm for 10 min and re-suspended in 30 mL Ni-NTA Binding Buffer (0.1 mol/L PBS, 0.5 mol/L NaCl, pH 8.0). The bacteria solution was lysed by sonication and clarified by centrifugation at 8000 rpm for 10 min at 4 °C. The resulting supernatant was further purified using Ni-NTA column (Cytiva, Stockholm, Sweden, 17526802) according to the manufacturer’s protocol. The protein concentration was measured spectrophotometrically, and the degree of protein purity was determined by SDS-PAGE.

### 2.3. Size Exclusion Chromatography (SEC)

Protein samples (TRIM21 and TRIMbody) were prepared at concentrations of 250 µg/mL in HEPES buffer. Each sample (500 µg) was injected onto an analytical Superdex^TM^ 200 Increase 10/300 GL column (Cytiva, Stockholm, Sweden, GE28-9909-44) connected to an FPLC ÄKTA BASIC pH/C system (GE Healthcare, Stockholm, Sweden, avant 150). HEPES (25 mM HEPES, pH 7.5, 200 mM NaCl) was used as the running buffer at the flow rate 0.5 mL/min, and the eluted proteins were monitored at 280 nm. A minimum of three independent experiments was performed. All proteins were stored in 20 mM Tris (pH 8.0), 150 mM NaCl, 1 mM DTT.

### 2.4. Binding ELISA

An enzyme-linked immunosorbent assay (ELISA) was used to determine the binding capability of the αEGFP TRIMbody to EGFP. EGFP and LaG16-LaG2 were expressed in *E. coli* BL21. Purified EGFP protein was coated on 96-well Costar half-area high-binding assay plates (Corning, Kennebunk, USA, 3690) overnight at 100 ng/well in PBS overnight at 4 °C, and blocked with 100 µL per well of 3% (*w/v*) blocking buffer (PBS with 3% BSA) at 37 °C for 1 h. The plates were washed with PBS with 0.05% Tween 20 (PBST), then threefold serial dilutions of αEGFP TRIMbody (RBCC-LaG16-LaG2), TRIM21 (RBCC-PRY-SPRY), αEGFP nanobody (LaG16-LaG2), and αHBsAg TRIMbody (RBCC-G12-scFv) were added and incubated at 37 °C for 1.5 h. Plates were washed five times with PBST and 50 µL of 1:1000 HRP conjugated anti-Flag antibody (Sigma-Aldrich, St. Louis, USA, A8592) in PBS were added per well before incubation at 37 °C for 45 min. After extensive washes with PBST, the binding activity was measured with the subsequent addition of diammonium 2,2′-azino-bis (3-ethylbenzothiazoline-6-sulfonate) (ABTS) substrate (Roche Applied Science, Mannheim, Germany, 11684302001) and the optical density of each well was read at 405 nm (OD405) using a Microplate Spectrophotometer (Biotek, Winooski, USA, Epoch).

### 2.5. Cell Culture and Transient Transfection

HEK293T cell lines were purchased from ATCC and maintained in Dulbecco’s modified Eagle’s medium (high glucose) supplemented with 10% fetal bovine serum (FBS) and 1% penicillin/streptomycin (10,000 units/mL) at 37 °C and 5% CO_2_. HEK293T cell lines were authenticated to be mycoplasma-negative using a Myco-Lumi™ Luminescent Mycoplasma Detection Kit (Beyotime, Shanghai, China, C0297S). HEK293T-EGFP cells were transient transfected with the appropriate plasmids using the Lipofectamine^TM^ 2000 Transfection Reagent (Invitrogen, Carlsbad, USA, 11668-019) according to the manufacturer’s instructions.

### 2.6. Generation of Cell Lines Stably Expressing EGFP and TRIMbody

Lentivirus was produced by co-transfection of HEK293T cells with a lentiviral transfer vector and packaging plasmids (psPAX2 and pMD2.G). Transfection was performed using PEIpro Transfection Reagent (Polyplus, Strasbourg, France, 115-0015) according to the manufacturer’s instructions. Cell culture supernatants were collected 48 h after transfection, filtered through a 0.45 μm filter, and added to trypsinized recipient cells (1 mL viral supernatant per well in a 6-well plate) supplemented with polybrene (8 μg/mL). The packaged viruses were used to transduce HEK293T-EGFP cells and HEK293T-EGFP/Tet-On-3G-αEGFP TRIMbody cells. HEK293T-EGFP-positive population of cells were sorted by flow cytometry from fluorescence-activated cell sorting (FACS) on a BD FACSAria Ⅱu using 405 nm lasers to isolate a low-level expressing EGFP-positive population of cells. HEK293T-EGFP cells engineered with the Tet-On-3G-αEGFP TRIMbody expression system were obtained by G418 selection (700 μg/mL) and then the surviving cells were seeded at 0.8 cells/well in 96-well plates for single-colony amplification. After 2 weeks in culture, single colonies were picked and split between two 48-well plates on separate plates. One half of monoclonal cell strains were first treated with 10 μg/mL of Doxycycline (Dox, Sigma Aldrich, St. Louis, USA, D9891) for 48 h to induce αEGFP TRIMbody expression and αEGFP TRIMbody-positive cells were selected by fluorescence microscopy screening. The other half of Tet-on-αEGFP TRIMbody-positive monoclonal cell strains were maintained and amplified using DMEM supplemented with 10% FBS without Dox addition for further characterization experiments.

### 2.7. Flow Cytometry Analysis

To analyze EGFP expression, cells were dissociated into single cells then added at a 1:2 *v/v* ratio to DMEM with 10% FBS. Data were collected on a BD FACSCalibur flow cytometer with a 488 nm laser for excitation and detection in the FITC channels and analyzed using FlowJo V10. FACS gating was based on the corresponding untreated cell.

### 2.8. Induction of αEGFP TRIMbody Expression with Doxycycline

293T-EGFP/Tet-On-3G-αEGFP TRIMbody cells were plated at a density of 30% confluence on plates in DMEM. After 24 h of seeding once the colonies have attached, 10 μg/mL of Dox was added to the medium and cells were cultured for 72 h replenishing with fresh Dox-containing medium every 24 h. To test for induction of αEGFP TRIMbody expression, control and test cell lines were harvested at the indicated times for protein extraction or fixed for immunostaining.

### 2.9. Laser Scanning Confocal Microscope and Live Cell Imaging

A laser scanning confocal microscope (Leica, Wetzlar, Germany, TCS-SP8) equipped with a 60× phase contrast oil immersion objective (numerical aperture = 1.0) was used to monitor the distribution and alteration of EGFP fluorescent signals from 293T-EGFP/Tet-on-αEGFP TRIMbody cells after treating with Dox. 293T-EGFP/Tet-On-αEGFP TRIMbody cells were cultured on 15 mm glass bottom culture dishes (Nest, Wuxi, China, 801002) and plated at a density of 30% confluence on plates in DMEM medium. After 24 h of seeding, 10 μg/mL of Dox was added to the medium and cells were cultured for 72 h replenishing with fresh Doxycycline-containing medium every 24 h. All live cell imaging was carried out on a DeltaVision Elite high-resolution cell imaging system (GE Healthcare), equipped with a 60× phase contrast oil immersion objective and live cell imaging environment control system (Live Cell Instrument). Approximately 3 × 10^5^ 293T-EGFP/Tet-on-αEGFP TRIMbody cells were seeded into each well of 4-chamer 35mm glass bottom dish with a 20 mm microwell (Cellvis, Hangzhou, China, D35C4-20-1-N) in the presence or absence of Dox. The chamber was supplemented with 5% CO_2_ and maintained at 37 °C with a microscope stage heater. After 24 h of seeding, scattering distributed 293T-EGFP/Tet-on-αEGFP TRIMbody cells were filmed for 6 h. Time series images of EGFP fluorescence were captured in 15 min intervals and then merged for visualization by softWoRx 6.5.

### 2.10. Immunostaining

Glass coverslips was put into 6-well plate, and 3 × 10^5^ cells were seeded on coverslips. Cells were fixed with 4% paraformaldehyde for 15 min at room temperature and then permeabilized with 0.3% Triton-X 100 and then blocked with 3% BSA for 1 h at room temperature. Post blocking, cells were incubated with the anti-Flag primary antibody (Yeasen, Shanghai, China, 30503ES20) overnight at 4 °C, then washed with PBS at room temperature and incubated with secondary antibody conjugated to Alexa^®^ Fluor 594 (Yeasen, Shanghai, China, 33212ES60) for 1 h at room temperature. The cells were washed and stained using Hoechst 33,342 (Yeasen, Shanghai, China, 40731ES10) to visualize the nuclei. A laser scanning confocal microscope (Leica, Wetzlar, Germany, TCS-SP8) was also used to investigate the colocalization of EGFP and αEGFP TRIMbody; this instrument is equipped with a 405 nm violet laser, a 488 nm blue laser, a 561 nm green laser, and a 639 nm red laser.

### 2.11. Protein Extraction and Western Blot Assay

For Western blot analysis, cells at the indicated times were pelleted, washed with 1×PBS, then lysed with RIPA buffer supplemented with protease inhibitor cocktail (Beyotime, Shanghai, China, P1010), phosphatase inhibitor cocktail (Beyotime, Shanghai, China, P1050), and 0.1% benzonase nuclease (Beyotime, Shanghai, China, D7121) on ice for 30 min and clarified by centrifugation at 10,000 rpm for 20 min at 4 °C and the supernatant fractions were collected. Total protein concentration was estimated using BCA protein assay kit (Pierce, Rockford, USA, 23227) and equivalent amounts (10 μg) of lysate were electrophoresed on 12% SDS-PAGE gel. Color Prestained Protein Standard was used to determine molecular weight. The gel was electro-blotted onto PVDF membrane (Merck Millipore, Carrigtwohill, Ireland, ISEQ00010) and blocked in TBS-T with 5% non-fat dried milk for 1 h at room temperature with gentle shaking. Membranes were incubated with primary antibodies at 4 °C overnight with gentle shaking, then washed three times with PBS-T. The membrane was then incubated with appropriate HRP-conjugated secondary antibodies in blocking buffer (TBS) for 1 h at room temperature with gentle shaking. Membranes were washed three times with PBS-T, blots were developed with enhanced chemiluminescence (ECL), and signals were captured with the chemiluminescence imaging system (Tanon, Shanghai, China, 5200).

### 2.12. RNA Isolation/cDNA Synthesis and Quantitative Real-Time PCR (qRT-PCR) Assay

Total RNA was extracted from cells by TRIzol reagent (Life Technologies, Austin, USA, 15596-026) following the manufacturer’s instructions. Ten micrograms of total RNA were converted to cDNA by performing reverse transcription PCR (RT-PCR) using PrimeScript™ RT reagent kit (TaKaRa, Dalian, China, RR037A). β-actin was used as internal reference genes for normalization. Quantitative real-time PCR was performed using a CFX Connect Real-Time PCR system (Bio-Rad) with a TB Green^®^ Premix Ex Taq™ Kit (TaKaRa, Dalian, China, RR420A) using the following protocol: pre-denaturation at 95 °C for 2 min; followed by 40 cycles of 5 s at 95 °C for denaturation and 30 s at 60 °C for annealing. At the end of the PCR cycles, melting curve analysis was performed to validate the specificity of the PCR products generated for each set of primers. Three technical replicates of each cDNA sample were collected. The primer sequences used for cDNA amplification (5′~3′) are listed in Appendix A. The relative quantification method (2^−ΔΔCT^) was used to evaluate quantitative variation between replicates examined.

### 2.13. Statistical Analysis

Statistical analyses were performed using Prism software (Version 8, GraphPad software). Error bars depict the SD or SEM as indicated. Statistical significance was calculated using an unpaired, two-tailed Student’s t test and depicted at the levels of * *p* < 0.05, ** *p* < 0.01, and *** *p* < 0.001.

## 3. Results

### 3.1. αEGFP TRIMbody has High Binding Activity to EGFP Protein In Vitro

TRIM21 belongs to the TRIM protein family and consists of a classic N-terminal RBCC motif and C-terminal PRY-SPRY domain. Among them, the RBCC motif can target protein to the proteasome via its E3 ubiquitin ligase activity and PRY-SPRY domain mediates immunoglobulin Fc fragment binding in a pincer-like interaction [32,33,34] (Figure 1a). To generate a new construct with both protein degradation activity and antibody-binding specificity, we fused the RBCC motif of TRIM21 to a nanobody that can bind to a targeted antigen. The corresponding protein was designated as TRIMbody, and the system was described as TRIMbody-Away. As shown in Figure 1b, TRIMbody is a multi-domain protein consisting of an N-terminal RING domain with E3 ubiquitin ligase activity, a B-box domain, a coiled-coil dimerization domain, and a C-terminal nanobody fragment which specifically recognizes intracellular proteins of interest.

To test whether TRIMbody could mediate degradation of the target protein, we chose EGFP protein as a proof-of-concept substrate. The previously reported nanobodies against EGFP, LaG16 and LaG2 were selected [35,36] and designed for bispecific anti-EGFP (αEGFP) nanobody. Then, an αEGFP TRIMbody was generated that comprised of an RBCC domain and bispecific αEGFP nanobody (Figure 1c). αEGFP TRIMbody could also be expressed as a soluble form in *E. coli* with the help of Lipoyl, as well as TRIM21. By analysis of SDS-PAGE, purified TRIM21 and αEGFP TRIMbody proteins exhibited major bands with molecular weights of 66 and 77 kDa, respectively (Figure 1d). Next, we examined the oligomeric state of the αEGFP TRIMbody by SEC analysis. The elute volume for αEGFP TRIMbody was determined at 10.3 mL, and 10.4 mL for purified TRIM21 protein, confirming that the αEGFP TRIMbody and TRIM21 have been correctly expressed (Figure 1e). Further, we measured the binding ability of αEGFP TRIMbody to EGFP by ELISA. Furthermore, a single-chain fragment of G12 antibody that recognizes HBsAg of HBV was used to generated αHBsAg TRIMbody and served as a negative control. We observed that the bispecific anti-EGFP nanobody (LaG16-LaG2) has a EC_50_ value of 2.40 nM in EGFP binding, while the αEGFP TRIMbody showed more evident binding activity to EGFP, with an EC_50_ value of 0.63 nM (Figure 1f), which could be due to dimerization of TRIM21 leading to enhanced binding avidity. Remarkably, there was no obvious EGFP binding activity for negative controls.

### 3.2. Degradation of Intracellular EGFP by Inducible Expression of αEGFP TRIMbody

Based on the EGFP binding by αEGFP TRIMbody, we next investigated its target protein degradation function in EGFP-expressing cells. To obtain the stably expressing EGFP cell lines, we performed live cell FACS sorting of 293T cells infected with lentivirus to isolate EGFP-positive cells, and the sorted cells were kept growing and still showed that 99.9% of the population are EGFP-positive at least for five passages, suggesting that expression of EGFP in 293T cells is relatively stable (Figure 2a). Expressing vectors with αEGFP TRIMbody were transiently transfected into EGFP stably expressing 293T cells, and then EGFP fluorescence intensity of cells at the indicated time was determined and analyzed via flow cytometry analysis. Meanwhile, vector, RBCC only, TRIM21 only, αEGFP nanobody, and αHBsAg TRIMbody were used as the controls. We observed that transfection of αEGFP TRIMbody resulted in significant decrease in EGFP fluorescence at 24 and 48 h in a time-dependent manner, but the fluorescence has no significant decrease at 72 h compared with 48 h, which may result from dilution loss of transfected αEGFP TRIMbody plasmid due to transient transfection, leading to insufficient expression of αEGFP TRIMbody protein in cells, or compensatory supply of EGFP protein by constitutive expression of EGFP. In contrast, transfection of RBCC, TRIM21, and αEGFP nanobody showed no decrease of the fluorescence (Figure 2b), suggesting that the RBCC domain and αEGFP nanobody failed to trigger intracellular EGFP degradation. Moreover, we found that αHBsAg TRIMbody, which was confirmed to have no EGFP binding ability in ELISA (Figure 1f), also did not induce intracellular EGFP degradation in EGFP-expressing 293T cells (Figure 2b). The results showed that the combination of the RBCC domain and specific antibody are necessary for degradation of intracellular EGFP.

Additionally, to avoid instability and side effects of transient transfection, the inducible expression of αEGFP TRIMbody by Doxycycline (Dox) treatment in stably expressing EGFP cells was constructed using the Tet-On-3G system and further degradation of EGFP was measured by confocal laser scanning microscope or flow cytometry. This system was sensitive to Dox treatment and the optimal Dox concentration was 10 μg/mL (Appendix A). Fluorescence images of cells showed that EGFP was universal and stably expressed and mainly located in the cytoplasm. Notably, αEGFP TRIMbody was not expressed in cells under no Dox treatment, while addition of Dox caused αEGFP TRIMbody expression that co-localized with EGFP (Figure 2e), suggesting that αEGFP TRIMbody can be successfully induced and bound to EGFP via the specificity of the anti-EGFP nanobody. Following Dox addition, EGFP aggregated quickly in cytosol with a spotty pattern at 24 h. In contrast, no significant difference of EGFP was observed without Dox treatment (Figure 2c). Moreover, the quantitative EGFP fluorescence of cells after Dox induction was measured to assess the efficacy of induced αEGFP TRIMbody for EGFP degradation. Cells were treated with Dox and then fixed in 4% paraformaldehyde, and the EGFP fluorescence was analyzed via flow cytometry. The EGFP fluorescence was decreased by 25% at 24 h and had a 67% reduction after 48 and 72 h (Figure 2d). Next, we examined the EGFP and αEGFP TRIMbody protein level after Dox treatment by Western blot analysis and found that αEGFP TRIMbody proteins in cells were induced and kept stably expressed at 24, 48, and 72 h (Figure 2f,g). As for EGFP protein levels, no significant difference was observed in the absence of Dox. By contrast, substantial degradation of EGFP in Dox-treated groups was observed, with 40% reduction at 24 h and 60% decrease at 48 and 72 h after Dox addition (Figure 2g), indicating that intracellular EGFP degradation was subject to Dox induction and αEGFP TRIMbody expression.

### 3.3. Dynamic Examination of EGFP Degradation and Accompanying Fluorescent Puncta within 24 h after αEGFP TRIMbody Induction

Due to the fact that there was a significant decrease of EGFP protein level at 24 h induction by Dox, we expected that the αEGFP TRIMbody-mediated EGFP degradation process may happen earlier. Therefore, to further characterize the EGFP degradation dynamic pattern and visualize EGFP fluorescence puncta over time upon αEGFP TRIMbody expression, cells were exposed to Dox or vehicle for as long as 24 h and imaged every 6 h using a laser scanning confocal fluorescence microscope (Figure 3a). Meanwhile, we quantified the EGFP fluorescence intensity of cells by flow cytometry analysis (Figure 3b). The punctate EGFP was observed 6 h later by Dox induction and the EGFP fluorescence signal was significantly decreased after 12 h of Dox treatment, reaching maximal reduction of 30% of EGFP fluorescence intensity compared to control at 24 h (Figure 3a,b). Next, EGFP fluorescent puncta was examined from fluorescent images using ImageJ software and we observed it appear as early as 6 h in Dox-treated cells (Figure 3c). Additionally, the puncta area increased dramatically by at least 60-fold compared to controls at 12 and 24 h of Dox treatment (Figure 3d). We speculate that formation of EGFP puncta upon Dox treatment may be due to of EGFP in cytosol through αEGFP TRIMbody induction and recruitment.

Furthermore, we assessed the EGFP and αEGFP TRIMbody protein level by Western blot analysis and found that EGFP significantly reduced at 24 h of Dox treatment, but not for 6, 12, and 18 h of Dox induction (Figure 3e,f). Meanwhile, αEGFP TRIMbody was successfully induced at 6 h of Dox treatment, reaching to higher levels after adding Dox for 12 to 24 h. We noticed that the EGFP puncta appeared at 6 h of Dox treatment, accompanied with induction of αEGFP TRIMbody, and followed by the reduction of the EGFP fluorescence signal. Finally, the EGFP protein level was detected to decrease at 24 h of induction. These results implied that EGFP degradation happened with the formation of puncta, which may be through αEGFP TRIMbody-mediated intracellular EGFP aggregation, then followed by change of conformation of EGFP, leading to reduction of the fluorescence signal, finally destructed by the RBCC domain-involved degradation pathway. In addition, we performed live-cell imaging on cells and observed the dynamic process of degradation of intracellular EGFP during αEGFP TRIMbody induction (Appendix A).

### 3.4. TRIMbody Induces Intracellular Protein Degradation through the Proteasome and Lysosomal Pathway

As mentioned above, the RBCC domain of TRIM21 has E3 ligase activity. To test whether αEGFP TRIMbody-induced EGFP degradation relied on proteasome activity, we used MG132, a proteasome inhibitor, to treat cells for 24 h with Dox induction, then cells were harvested for imaging and assessed for EGFP fluorescence intensity as described above. The results showed that the amount of EGFP puncta was less and EGFP fluorescence intensity was stronger in the presence of MG132 (Dox+MG132+) compared to no MG132-treated group (Dox+) (Figure 4a,b), indicating that inhibition of the proteasome degradation pathway by MG132 treatment contributed to suppression of αEGFP TRIMbody-mediated intracellular protein degradation. However, the EGFP fluorescence intensity in the MG132-treated group (Dox+MG132+) was still not comparable with the group without Dox treatment (Dox−) (Figure 4b), suggesting that another degradation pathway may be involved. Besides the ubiquitin proteasome system, intracellular proteins could be degraded via the lysosomal pathway. To further test if EGFP protein degradation by αEGFP TRIMbody was mediated through the lysosome, we treated cells with the autophagy–lysosome inhibitor Chloroquine (CQ, 40 μM). Strikingly, Chloroquine treatment reduced the EGFP puncta area, partially rescuing the EGFP fluorescence intensity reduction by Dox treatment (Figure 4c,d). In addition, Chloroquine treatment had a more obvious rescue effect compared to MG132 treatment, with less EGFP puncta and higher EGFP fluorescence intensity, suggesting that the autophagy–lysosome pathway contributed more to αEGFP TRIMbody-mediated EGFP degradation. We also noticed that the relative mRNA levels of ubiquitin B, MAP1LC3A, MAP1LC3B, ULK1, SQSTM1/p62, Atg5, Beclin1/Atg6, Atg7, and Atg12 were increased upon overexpression of αEGFP TRIMbody (Appendix A). Based on these results, we concluded that both MG132 and Chloroquine treatment could, but not completely, rescue αEGFP TRIMbody-mediated intracellular EGFP degradation. Therefore, TRIMbody-Away technology could be utilized for intracellular protein degradation and relies on both proteasome and autophagy–lysosome pathways.

## 4. Discussion

Examining gene function in different cell types or tissues exists in at least three layers of perturbation, including DNA modification, RNA interference, and protein degradation [37,38]. However, the utility of DNA/RNA editing methods can be limited by reducing of the target protein through a long time of process including DNA/RNA-targeted excision and protein turnover, which may delay the manifestation of phenotypes and activate a compensatory mechanism. In contrast, techniques for disrupting intracellular protein enable the direct analysis of its biological function. Recently, TRIM-Away, a promising approach to degrade endogenous proteins acutely and rapidly in mammalian cells, was developed to remove unmodified native proteins by microinjection of anti-targets antibodies and TRIM21 protein into cells. However, the difficulty on manipulations of a bulk cell population limited its extensive application. In addition, it was reported that TRIM21 is involved in the regulation of innate immunity and the inflammatory IFN pathway [39,40]. Thus, exogenous induction of TRIM21 by microinjection or electrotransfection needs rigorous investigation. The relatively large size of the full-size mAbs (150 kDa) results in correspondingly low tissue penetration and inaccessibility of some sterically hindered epitopes, which further limits the degrading efficacy of endogenous protein by Trim-Away Technology. In order to resolve the concern of potential side effects of full-size mAbs and TRIM21, we established an alternative approach by fusing of the function RBCC domain of TRIM21, containing E3 ubiquitin ligase motif, with the fragment of anti-EGFP nanobody, to target intracellular EGFP for degradation.

Next, by transient transfection and Tet-on inducible expression of αEGFP TRIMbody in stable EGFP-expressing HEK293T, we revealed the intracellular EGFP degradation mediated by αEGFP TRIMbody in a time-dependent pattern. Moreover, by examination of EGFP puncta, EGFP fluorescence intensity, and protein level upon αEGFP TRIMbody at different time points of Dox induction, we observed EGFP puncta firstly appeared, followed by a decrease of fluorescence intensity, and finally destruction/degradation of EGFP protein, implying a dynamic degradation process of intracellular EGFP regulated by αEGFP TRIMbody. The different period of time of EGFP fluorescence intensity deduction and EGFP protein degradation indicated that EGFP fluorescence intensity loss was not fully represented by protein degradation, and might through an intermediate state, may probably be due to change of protein confirmation. Thus, the calculation of EGFP degradation time of TRIMbody-Away according to fluorescence intensity loss was definitely worth negotiating over. Furthermore, HEK293T-Low-EGFP cells engineered with the Tet-On-3G-αEGFP TRIMbody expression system were also obtained as described above. Expressing αEGFP TRIMbody in stable HEK293T-Low-EGFP cells could degrade intracellular EGFP in a similar time-dependent manner (Appendix A). In conclusion, the Tet-on inducible TRIMbody system has no side effects of transfection, is easy to handle by addition of Dox to culture media or washing it out to reverse the degradation effect, and potentially could be used to dynamically observe the relevant phenotypes associated with target protein degradation.

The majority of cellular proteins are rapidly degraded and compensated with newly synthesized copies [41]. Thus, exploring the function of long-lived intracellular proteins is more challenging [42,43,44,45]. In this study, we used EGFP as a model substrate, which has a long life time and is hard to turnover, and also EGFP expression was stably and constitutively driven by the EF promoter in a lentivirus construct, and these reasons might explain why αEGFP TRIMbody-mediated intracellular EGFP was not completely degraded in our observation. Another possibility was that the inducible expression of αEGFP TRIMbody could only be detected at 6 h post Dox treatment with few molecules, which might not be sufficient for disrupting the intracellular EGFP protein. Although αEGFP TRIMbody showed promising intracellular protein degradation ability in cytoplasm, we did not evaluate the degradation ability for an endogenous native protein. It is worthy of further exploration as the growing pool of nanobodies directly recognizing endogenous proteins are available.

## 5. Conclusions

By fusing a nanobody with the RBCC motif of TRIM21, a novel fusion protein that specifically degraded intracellular protein was generated, and this protein was termed as TRIMbody. We introduced EGFP as a model substrate and generated αEGFP TRIMbody using a bispecific anti-EGFP (αEGFP) nanobody. Next, by inducible expression of αEGFP TRIMbody in stable HEK293T-EGFP cells, we demonstrated this system could degrade intracellular EGFP in a time-dependent manner. Further, addition of proteasome inhibitor and autophagy–lysosome inhibitor suppressed the degradation of intracellular EGFP protein, demonstrating that EGFP protein degradation mediated by αEGFP TRIMbody relies on both the proteasome and autophagy–lysosome pathways. Thus, TRIMbody may be used as a powerful strategy for degrading intracellular proteins. In addition, as the CRISPR-Cas9-mediated knock-in method becomes increasingly popular, in situ tagging with GFP is likely to become commonplace and it is worthy to try this system for degradation of GFP-tagged endogenous proteins. Moreover, by using cell-type/tissue-specific promoter to drive expression of the αEGFP TRIMbody transgene, it is possible to try this system in a cell-type/tissue-specific degradation in vivo. Collectively, TRIMbody-Away technology could be exploited to specifically degrade intracellular protein and may expand the potential applications of degrader technologies.

## Figures and Tables

**Figure 1 biomolecules-11-01512-f001:**
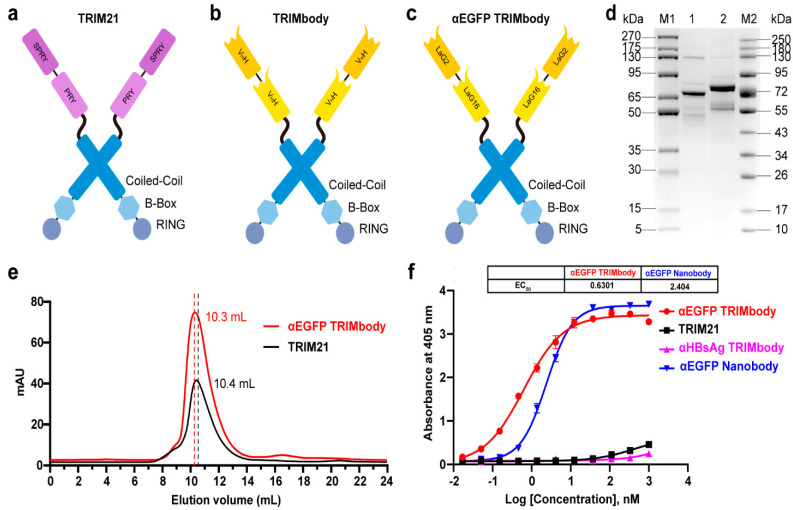
Characterization of TRIM21 and TRIMbody. (**a**,**b**) Schematic representation of TRIM21 (**a**) consisting of an N-terminal RBCC motif and a C-terminal PRY-SPRY domain, and TRIMbody (**b**) consisting of an RBCC motif and a bispecific nanobody. (**c**) Schematic representation of the αEGFP TRIMbody consisting of an N-terminal RBCC motif and a C-terminal bispecific anti-EGFP (αEGFP) nanobody (LaG16-LaG2). (**d**) Coomassie-stained gel shows TRIM21 and αEGFP TRIMbody proteins. Lane 1, protein molecular weight marker 1 (5~270 kDa); Lane 2, TRIM21; Lane 3, αEGFP TRIMbody; Lane 4, protein molecular weight marker 2 (10~250 kDa). (**e**) Size exclusion chromatography of αEGFP TRIMbody and TRIM21. Protein samples were loaded onto an analytical Superdex^TM^ 200 Increase 10/300 GL column connected to an FPLC ÄKTA BASIC pH/C system (GE Healthcare). (**f**) The binding activities of αEGFP TRIMbody, TRIM21, αEGFP Nanobody, and αHBsAg TRIMbody to EGFP were evaluated by ELISA. The EGFP were coated on ELISA plates, and HRP-conjugated anti-Flag antibody was used for detection of binding TRIM21, αEGFP TRIMbody, αEGFP Nanobody, and αHBsAg TRIMbody. Data are shown as mean ± SD.

**Figure 2 biomolecules-11-01512-f002:**
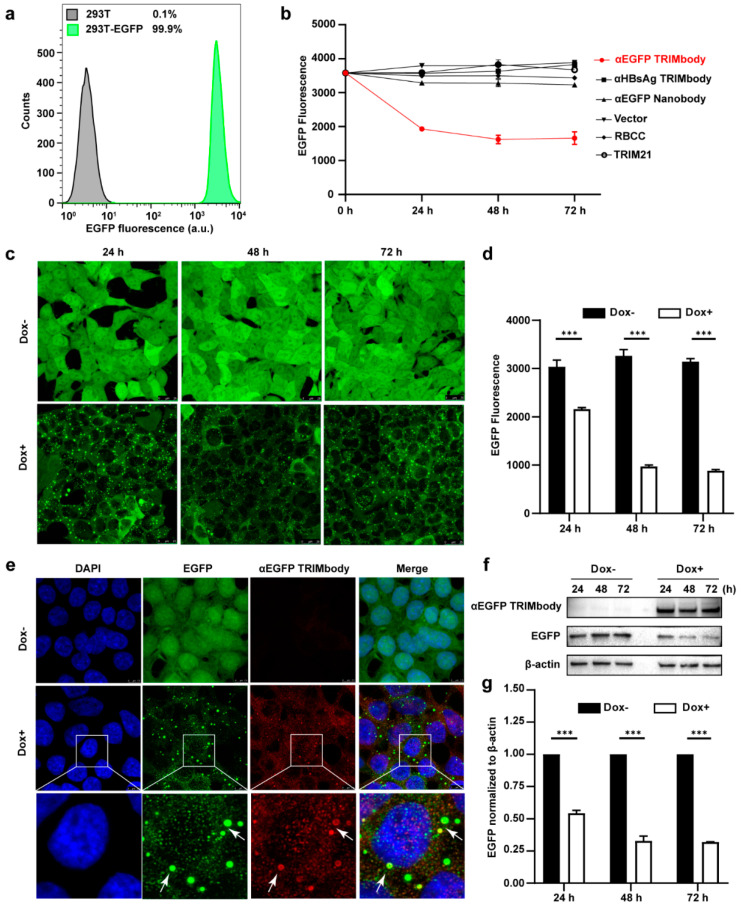
Degradation of EGFP by αEGFP TRIMbody. (**a**) Stable expressing EGFP 293T cell lines were analyzed by flow cytometry. At least 15,000 cells were counted for each experiment. Percentages correspond to EGFP-positive cells falling within the gate were drawn. (**b**) EGFP expressing was decreased in cells after αEGFP TRIMbody transfection. Error bars indicate standard deviations (*n* = 3). (**c**) The inducible expression of αEGFP TRIMbody by Dox treatment in stable HEK293T-EGFP cells caused EGFP degradation. Scale bars, 25 μm. (**d**) Mean fluorescence intensity of EGFP in cells untreated or treated with Dox. Mean fluorescence intensity was measured using flow cytometry and indicated by bar graphs (*n* = 3 replicates per group). Data represent the mean ± SEM. *** *p* < 0.001 represents statistical significance. (**e**) Subcellular localization of αEGFP TRIMbody and EGFP in αEGFP TRIMbody-inducible and EGFP-stable expressing cells. Nuclei were stained with Hoechst 33342; green represents the EGFP protein, red represents the αEGFP TRIMbody tagged with anti-Flag antibody conjugated-Alexa^®^ Fluor 594. White arrows indicate co-localization of EGFP and αEGFP TRIMbody in cytosol of the cells. Scale bars, 7.5 μm. (**f**,**g**) The degradation of EGFP was determined by Western blot analysis. Cells were treated with Dox for 72 h and equal amounts of cell lysates (10 μg) were loaded in each well. Through immunoblotting with anti-Flag antibody, the relative optical density of bands on the blots was analyzed by software. Values are the mean ± SEM (*n* = 3/group). Statistical significance between ligands were determined using a two-way ANOVA test. *** *p* < 0.001 versus control.

**Figure 3 biomolecules-11-01512-f003:**
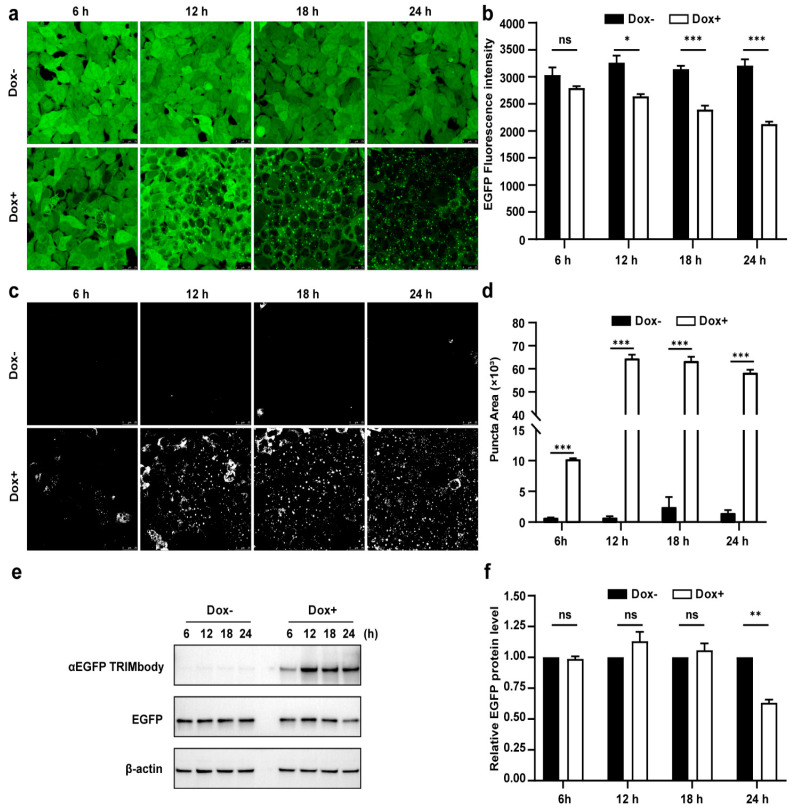
Characterization of αEGFP TRIMbody temporal expression pattern. (**a**) Laser scanning confocal fluorescence microscopy images showed the change in EGFP expression over time with and without Doxycycline. Scale bars are 25 µm. (**b**) Mean fluorescence intensity of EGFP in cells that were untreated or treated with Dox. Mean fluorescence intensity is measured using flow cytometry and indicated by bar graphs (*n* = 3 replicates per group). Data represent the mean ± SEM. The ns represents no significance, * represents *p* < 0.05, and *** represents *p* < 0.001. (**c**) EGFP fluorescent puncta was examined from fluorescent images using ImageJ software. (**d**) Relative EGFP puncta area of autophagosomes or autolysosomes was measured using ImageJ software. Statistical analysis of the puncta area of autophagosomes and autolysosomes per cell were samples from a pool of at least 3 images. Data represent the mean ± SEM. ns represents no significance, *** represents *p* < 0.001. (**e**) Western blotting analyzed EGFP and αEGFP TRIMbody levels in the total cell lysate at the indicated times. Equal amounts of cell lysates (10 μg) were loaded in each well and immunoblotted with anti-Flag antibody. (**f**) Graphs show statistic results from relative optical density of bands on the blots. Values are the mean ± SEM. (*n* = 3/group). Statistical significance between ligands was determined using a two-way ANOVA test. ** *p* < 0.01, versus control.

**Figure 4 biomolecules-11-01512-f004:**
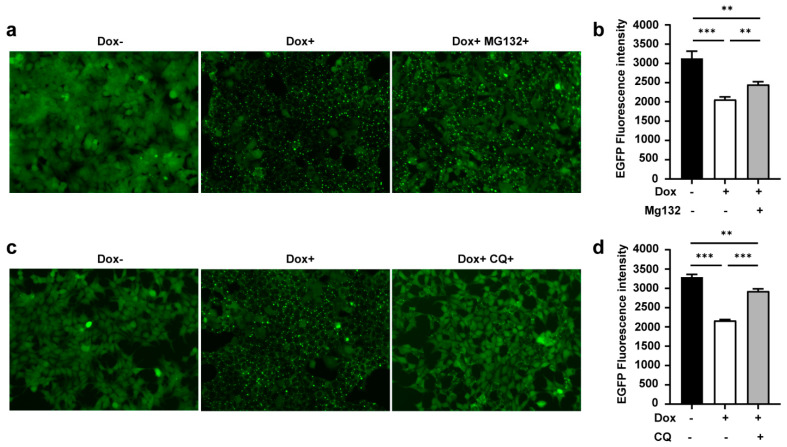
TRIMbody induced ubiquitination and proteasome-mediated degradation of EGFP. (**a**) Effect of proteasome inhibitor MG132 on degradation of EGFP. (**b**) Average fluorescence intensity of EGFP in cells untreated or treated with Dox or MG132 was measured using flow cytometry and indicated by bar graphs (*n* = 3 replicates per group). Data represent the mean ± SEM. ** *p* < 0.01 and *** *p* < 0.001 represent statistical significance. (**c**) Effect of autophagy–lysosome inhibitor Chloroquine (CQ) on degradation of EGFP. (**d**) Average fluorescence intensity of EGFP in cells untreated or treated with Dox or Chloroquine (CQ) was measured using flow cytometry and indicated by bar graphs (*n* = 3 replicates per group). Data represent the mean ± SEM. ** *p* < 0.01 and *** *p* < 0.001 represent statistical significance.

## Data Availability

Not applicable.

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
