# Peer review of "A Promising Intracellular Protein-Degradation Strategy: TRIMbody-Away Technique Based on Nanobody Fragment"

_biomolecules, 2021, doi:10.3390/biom11101512_

Round 1

Reviewer 1 Report

In this work, a novel fusion protein that specifically degraded intracellular protein was generated, which named as TRIM-body by fusing nanobody with RBCC motif of TRIM21. Through introducing anti-EGFP bispecific nanobody (αEGFP), the author designed αEGFP TRIMbody that specifically bind to EGFP with high affinity. Furthermore, expressing αEGFP TRIMbody in stable HEK293T-EGFP cells could degrade intracellular EGFP in a time-dependent manner. TRIMbody-Away technology could be utilized to specifically degrade intracellular protein and may expand the potential applications of degrader technologies. However, there are still some problems in the content, which are list in special comments. Overall, revision should be processed for improving article quality and publishing in this journal.

Specific comments:

  1. This research mentioned “purified TRIM21 and αEGFP TRIMbody proteins exhibited major bands with molecular weights of 66 kDa and 77 kDa ” , but in Figure 1D, the size of line1 and line2 was not clear, it is recommended to make the Figure 1d with more accurate protein molecular weight marker.
  2. The Y axis of figure 3b, figure 3d and figure 3f was not aligned.
  3. There are some sentences with repetitive meaning in the last part of abstract and conclusion, it is suggested to be concise.  

Author Response

Authors Response

Point-by-point responses to the reviewers’ comments:

In this work, a novel fusion protein that specifically degraded intracellular protein was generated, which named as TRIM-body by fusing nanobody with RBCC motif of TRIM21. Through introducing anti-EGFP bispecific nanobody (αEGFP), the author designed αEGFP TRIMbody that specifically bind to EGFP with high affinity. Furthermore, expressing αEGFP TRIMbody in stable HEK293T-EGFP cells could degrade intracellular EGFP in a time-dependent manner. TRIMbody-Away technology could be utilized to specifically degrade intracellular protein and may expand the potential applications of degrader technologies. However, there are still some problems in the content, which are list in special comments. Overall, revision should be processed for improving article quality and publishing in this journal.

Specific comments:

  1. This research mentioned “purified TRIM21 and αEGFP TRIMbody proteins exhibited major bands with molecular weights of 66 kDa and 77 kDa”, but in Figure 1D, the size of line1 and line 2 was not clear, it is recommended to make the Figure 1d with more accurate protein molecular weight marker.

Response: We greatly appreciate the Reviewer’s suggestion. To address this problem, we performed SDS-PAGE again using two types of color prestained protein standard marker. The protein standard markers M00624-250 from GenScript and P7719S from NEW ENGLAND BioLabs, covering a wide range of molecular weights, could indicate the TRIM21 protein with a molecular weight of 66 kDa and αEGFP TRIMbody with a molecular weight of 77 kDa. The original figure and revised figure are shown below.

Coomassie-stained gel shows TRIM21 and αEGFP TRIMbody. (a) Original figure. Lane 1, TRIM21; Lane 2, αEGFP TRIMbody; Lane 3, protein molecular weight marker (11~245 kDa). (b) Revised figure. Lane 1, protein molecular weight marker 1 (5~270 kDa); Lane 2, TRIM21; Lane 3, αEGFP TRIMbody; Lane 4, protein molecular weight marker 2 (10~250 kDa).

  1. The Y axis of figure 3b, figure 3d and figure 3f was not aligned.

Response: We thank the Reviewer for the valuable suggestion. In the revised manuscript, the Y axis of figure 3b, figure 3d and figure 3f have been aligned by using Adobe Illustrator software.

  1. There are some sentences with repetitive meaning in the last part of abstract and conclusion, it is suggested to be concise.

Response: We thank the Reviewer for the valuable comment. The conclusion has been revised in the text. As seen below:

“By fusing nanobody with RBCC motif of TRIM21, a novel fusion protein that spe-cifically degraded intracellular protein was generated, and this protein was termed as TRIMbody. Next, by inducible expression of αEGFP TRIMbody in stable HEK293T-EGFP cells, we demonstrated this system could degrade intracellular EGFP in a time-dependent manner. Further, addition of proteasome inhibitor and autophagy-lysosome inhibitor suppressed the degradation of intracellular EGFP protein, demonstrating that αEGFP TRIMbody relies on both the proteasome and autophagy-lysosome pathways. Thus, TRIMbody may be used as a powerful strategy for degrading intracellular proteins. In addition, as the CRISPR-Cas9-mediated knock-in method becomes increasingly popular, in situ tagging with GFP is likely to become commonplace and it is worthy to try this system for degradation of GFP tagged endogenous proteins. Moreover, by using cell-type/tissue specific promoter to drive expression of αEGFP TRIMbody transgene, it is possible to try this system in a cell-type/tissue specific degradation in vivo. Collectively, TRIMbody-Away technology could be exploited to specifically degrade intracellular protein and may expand the potential applications of degrader technologies.”

Reviewer 2 Report

Comments and suggestions to the authors:

This article is well written and interesting. I think this manuscript should be published. However, I have few minor comments and critics given below: 

1) Part 4 (Discussion) is too lengthy and very confusing. I would like to suggest authors, please rewrite this section in a very precise, concise and reader friendly manner. Discussion must be clear, easy to understand and coherent to readers. During writing this section, please ensure that the reader can easily follow your ideas and train of thoughts and learn something new from this manuscript.

2) Part 5, In conclusion section, please add some future plans in this scope of study, regarding its applicability to the routine clinical practice and how do you see to involve the suggested approach and methods.

Author Response

Authors Response

Point-by-point responses to the reviewers’ comments:

Comments and suggestions to the authors:

This article is well written and interesting. I think this manuscript should be published. However, I have few minor comments and critics given below: 

  • Part 4 (Discussion) is too lengthy and very confusing. I would like to suggest authors, please rewrite this section in a very precise, concise and reader friendly manner. Discussion must be clear, easy to understand and coherent to readers. During writing this section, please ensure that the reader can easily follow your ideas and train of thoughts and learn something new from this manuscript.

Response: Thank you for this valuable comment. As suggested by the Reviewer, we rewrote the discussion section. As seen below:

“Examining gene function in different cell types or tissues exist at least three layers of perturbation, including DNA modification, RNA interference and protein degradation [37,38]. However, the utility of DNA/RNA editing methods can be limited by reducing of the target protein through long time of process including DNA/RNA targeted excision and protein turnover, which may delay the manifestation of phenotypes and activate compensatory mechanism. In contrast, techniques for disrupting intracellular protein enable the direct analysis of its biological function. Recently, Trim-Away, a promising approach to degrade endogenous proteins acutely and rapidly in mammalian cells, was developed to remove unmodified native proteins by microinjection of anti-targets anti-bodies and TRIM21 protein into cells. However, the difficulty on manipulations of bulk cell population limited it extensive application. In addition, it was reported that TRIM21 involved in the regulation of innate immunity and the inflammatory IFN pathway [39,40]. Thus, exogenous induction of TRIM21 by microinjection or electrotransfection need rigorous investigation. In order to resolve the concern of potential side-effects of full-size TRIM21, we established an alternative approach by fusing of function RBCC domain of Trim21, containing E3 ubiquitin ligase motif, with the fragment of anti-EGFP nanobody, to target intracellular EGFP for degradation.

Next, by transient transfection and Tet-on inducible expression of αEGFP TRIMbody in stable EGFP expressing HEK293T, we revealed the intracellular EGFP degradation mediated by αEGFP TRIMbody in a time-dependent pattern. Moreover, by examination EGFP puncta, EGFP fluorescence intensity and protein level upon αEGFP TRIMbody induction at different time point of Dox induction, we observed EGFP puncta firstly appeared, followed by decrease of fluorescence intensity and finally destruction/degradation of EGFP protein, implying a dynamic degradation process of intracellular EGFP regulated by αEGFP TRIMbody. The different time period of EGFP fluorescence intensity deduction and EGFP protein degradation indicated that EGFP fluorescence intensity loss was not fully represented by protein degradation, and might through an intermediate state probably due to change of protein confirmation. Thus, the calculation of EGFP degradation time of TRIM-away according to fluorescence intensity loss was definitely worth negotiating over. Furthermore, HEK293T-Low-EGFP cells engineered with the Tet-On-3G-αEGFP TRIMbody expression system were also obtained as described above. Expressing αEGFP TRIMbody in stable HEK293T-Low-EGFP cells could degrade intracellular EGFP in a similar time-dependent manner (Figure S3). In conclusion, Tet-on inducible TRIMbody system has no side effect of transfection, easy to handle by addition of Dox to culture media or washing it out to reverse the degradation effect, and potentially could be used to dynamically observe the relevant phenotypes associated with target protein degradation.

The majority of cellular proteins are rapidly degraded and compensated with newly synthesized copies [41]. Thus, exploring function of long-lived intracellular proteins is more challenging [42-45]. In this study, we used EGFP as a model substrate which has long life time and hard to turnover, also EGFP expression was stably and constitutively driven by the EF promoter in lenti-virus construct, these reasons might explain why αEGFP TRIMbody mediated intracellular EGFP was not completely degraded in our observation. Another possibility was that the inducible expression of αEGFP TRIMbody could only be detected at 6 h post Dox treatment with few molecules, which might not be sufficient for disrupting intracellular EGFP protein. Although αEGFP TRIMbody showed promising intracellular protein degradation ability in cytoplasm, we did not evaluate the degradation ability for endogenous native protein. It is worthy of further exploration as the growing pool of nanobodies directly recognizing endogenous proteins are available.”

  • Part 5, In conclusion section, please add some future plans in this scope of study, regarding its applicability to the routine clinical practice and how do you see to involve the suggested approach and methods.

Response: We thank the Reviewer for the valuable comment. We have discussed this issue in the text. As seen below:

“By fusing nanobody with RBCC motif of TRIM21, a novel fusion protein that spe-cifically degraded intracellular protein was generated, and this protein was termed as TRIMbody. Next, by inducible expression of αEGFP TRIMbody in stable HEK293T-EGFP cells, we demonstrated this system could degrade intracellular EGFP in a time-dependent manner. Further, addition of proteasome inhibitor and autophagy-lysosome inhibitor suppressed the degradation of intracellular EGFP protein, demonstrating that αEGFP TRIMbody relies on both the proteasome and autophagy-lysosome pathways. Thus, TRIMbody may be used as a powerful strategy for degrading intracellular proteins. In addition, as the CRISPR-Cas9-mediated knock-in method becomes increasingly popular, in situ tagging with GFP is likely to become commonplace and it is worthy to try this system for degradation of GFP tagged endogenous proteins. Moreover, by using cell-type/tissue specific promoter to drive expression of αEGFP TRIMbody transgene, it is possible to try this system in a cell-type/tissue specific degradation in vivo. Collectively, TRIMbody-Away technology could be exploited to specifically degrade intracellular protein and may expand the potential applications of degrader technologies.”

Round 2

Reviewer 1 Report

I think the authors have well response to the comments of reviewers and suggest the paper should be accepted.